# Impact of the COVID-19 pandemic on the provision and uptake of services for the prevention of mother-to-child transmission of HIV in Zimbabwe

**Elizabeth Chappell**[1]*, **Anesu Chimwaza**[2], **Ngoni Manika**[2], **Catherine J. Wedderburn**[1,3], **Zivai Mupambireyi Nenguke**[4], **Hannah Gannon**[5], **Frances Cowan**[4,6], **Tom Gibb**[7], **Michelle Heys**[5], **Felicity Fitzgerald**[8], **Andrew Phillips**[9], **Simbarashe Chimhuya**[10], **Diana M. Gibb**[1], **Deborah Ford**[1], **Angela Mushavi**[2], **Mutsa Bwakura-Dangarembizi**[10]

**1** MRC Clinical Trials Unit at UCL, London, United Kingdom, **2** Ministry of Health and Child Care, Harare, Zimbabwe, **3** Department of Paediatrics and Child Health, University of Cape Town, Cape Town, South Africa, **4** Centre for Sexual Health and HIV/AIDS Research (CeSHHAR), Harare, Zimbabwe, **5** UCL Great Ormond Street Institute of Child Health, University College London, London, United Kingdom, **6** Liverpool School of Tropical Medicine, Liverpool, United Kingdom, **7** Picturing Health, United Kingdom, **8** Department of Infectious Disease, Imperial College London, London, United Kingdom, **9** Institute for Global Health, UCL, London, United Kingdom, **10** Child and Adolescent Health Unit, Faculty of Medicine and Health Sciences, University of Zimbabwe, Harare, Zimbabwe

* e.chappell@ucl.ac.uk

**Data Availability Statement:** The data underlying this analysis are available as a supplementary file.

## Abstract

Zimbabwe is targeting elimination of mother-to-child transmission of HIV by December 2025, however the COVID-19 pandemic challenged health service delivery globally. Monthly aggregated data were extracted from DHIS-2 for all facilities delivering antenatal care (ANC). ZIMSTAT and Spectrum demographic estimates were used for population-level denominators. Programme indicators are among those in HIV care and population indicators reflect the total population. The mean estimated proportion of pregnant women booking for ANC per month did not change (91% pre-pandemic vs 91% during pandemic, p = 0.95), despite dropping to 47% in April 2020. At a programme-level, the estimated proportion of women who received at least one HIV test fell in April 2020 (3.6% relative reduction vs March (95% CI 2.2–5.1), p<0.001) with gradual recovery towards pre-pandemic levels. The estimated proportion of women who were retested among those initially negative in pregnancy fell markedly in April 2020 (39% reduction (32–45%), p<0.001) and the subsequent increase was much slower, only reaching 39% by September 2021 compared to average 53% pre-pandemic. The mean estimated proportion of pregnant women with HIV on ART was unchanged at programme-level (98% vs 98%, p = 0.26), but decreased at population-level (86% vs 80%, p = 0.049). Antiretroviral prophylaxis coverage decreased among HIV-exposed infants, at programme- (94% vs 87%, p = 0.001) and population-levels (76% vs 68%, p<0.001). There was no significant change in HIV-exposed infants receiving EID (programme: 107% vs 103%, p = 0.52; population: 87% vs 79%, p = 0.081). The estimated proportion of infants with HIV diagnosed fell from 27% to 18%, (p<0.001), while the estimated proportion on ART was stable at a programme (88% vs 90%, p = 0.82) but not

**Funding:** This project was funded by ViiV Healthcare (funding received by principal investigator MBD). The MRC Clinical Trials Unit at UCL is supported by the Medical Research Council (programme number MC_UU_00004/03). FF is supported by an NIHR Development and Skills Enhancement Award. The funders had no role in study design, data collection and analysis, decision to publish, or preparation of the manuscript.

**Competing interests:** The authors have declared that no competing interests exist.

population (22% vs 16%, p = 0.004) level. Despite a drop at the start of the pandemic most programme indicators rapidly recovered. At a population-level indicators were slower to return, suggesting less women with HIV identified in care.

## Introduction

The COVID-19 pandemic and associated control measures challenged health service delivery globally. Governments took unprecedented measures to limit the spread of virus, including stay-at-home orders and school closures. Healthcare services were overburdened by rising COVID-19 caseloads, the reallocation of staff and other resources, and supply chain bottle-necks, while patients experienced barriers in accessing care due to fear of contracting COVID-19, closure of public transport services and financial difficulties [1].

In Zimbabwe the first confirmed case of COVID-19 was on the 21st March 2020, and on the 30th March a national lockdown was imposed initially for 21 days, with measures including orders to stay at home unless an essential worker or for food or healthcare, and the suspension of public transport [2]. National control measures continued for much of the year, and similar restrictions were introduced during subsequent waves of the pandemic in January and June 2021. There have been two recent periods of healthcare worker strikes in Zimbabwe; a doctors' strike pre-pandemic from 3rd September 2019 to 22nd January 2020, and a nurses' strike from 17th June 2020 to 9th September 2020, which was held partly in response to problems exacerbated by the pandemic including a lack of personal protective equipment (PPE) [3].

An impact on maternal and child health services has been observed in other pandemics. During the Ebola pandemic in Guinea and Sierra Leone key health indicators deteriorated, including number of antenatal care (ANC) visits and infant vaccination, and these were slow to return to normal following the end of pandemic [4, 5]. This impact demonstrates the importance of protecting the populations of pregnant women and their infants; women living with HIV are particularly vulnerable. Early in the COVID-19 pandemic, modelling studies estimated a 6 month interruption to ART supply for 50% of people in sub-Saharan Africa would lead to a 1.63-fold increase in death and 1.64-fold increase in mother-to-child transmission (MTCT) over one year [6]. Recent data reported by PEPFAR from 19 African countries (including Zimbabwe) demonstrated a fall in the number of individuals initiating ART between October 2019 and March 2021, with number of children <15 years of age slower to recover [7].

The antenatal seroprevalence of HIV in Zimbabwe was estimated to be 11.6% in 2021, and the country has made noteworthy progress in identifying and treating pregnant women living with HIV in recent years, aiming to achieve elimination of MTCT by December 2025 [8]. In 2018, antenatal HIV testing coverage was 98%, and an estimated 93% of all pregnant and breastfeeding women living with HIV received antiretroviral therapy (ART). The risk of MTCT fell from 22% in 2010 to 7.8% in 2018. During 2019, 56% of HIV-exposed infants received early diagnosis for HIV (EID). In response to the pandemic, the Ministry of Health and Child Care identified essential healthcare services, which included ANC and prevention of MTCT (PMTCT) services, and developed national guidelines on delivery of HIV care services during the pandemic. Materials were developed to support community healthcare workers and the population, including radio question and answer sessions with women on the topics of ANC, HIV and COVID-19.

In this analysis we explore the impact of the COVID-19 pandemic on the provision and uptake of services for PMTCT of HIV across Zimbabwe.

## Methods

### Ethics statement

The study received ethics approval from the Medical Research Council of Zimbabwe (MRCZ/A/2682). Individual level informed consent was not required, given the use of aggregate routine Ministry of Health and Child Care data.

### Data and methods

Several sources of data were used to estimate key PMTCT indicators. Firstly, aggregate data including measures relating to PMTCT, were obtained from the District Health Information System (DHIS-2) for all 1560 healthcare facilities in Zimbabwe which provide ANC (S1 File) [9]. Population-level denominators were obtained from ZIMSTAT (numbers of pregnant women) and the Spectrum/AIDS Impact Model (AIM) (numbers of pregnant women with HIV; numbers of HIV-exposed infants; numbers of HIV-infected infants) [10, 11]. The Spectrum/AIM model estimates key HIV indicators, including those relating to PMTCT; estimates are calculated each year by the Ministry of Health and Child Care. Spectrum estimates for 2021 were not available at the time of analysis, and were assumed to be the same as 2020. ZIMSTAT data come from national census reports. All denominators from Spectrum and ZIMSTAT were estimated per year and divided by 12 to give monthly estimates.

Eleven key PMTCT indicators for services were estimated, using definitions described in Table 1; two related to ANC, four to maternal HIV care in ANC, three to care for HIV-exposed infants, and two to care for infants with HIV. Where relevant, two types of indicators were considered: programme estimates were among those engaged in HIV care; and population indicators for the total population. Indicators were estimated at both a national and provincial level if using ZIMSTAT denominators, and a national level only for indicators using Spectrum denominators (as these estimates are not available by province).

Indicators were compared before and during the pandemic. Before the pandemic was defined as January 2017-March 2020 (excluding September 2019-January 2020 during the doctors' strike) and during the pandemic as April 2020-September 2021. Months covering the nurses' strike (July-August 2020) were not excluded from the primary analysis given that the strike was in part a response to the pandemic. Two sensitivity analyses were conducted: the first excluded months during the nurses' (as well as doctors') strike from analysis; the second included months during both strike periods. The coverage of EID (indicator 9) was only reported to June 2020 as guidelines changed from July 2020 to recommend testing HIV-exposed infants at birth as well as six weeks, making comparisons over later time periods unreliable. Healthcare facility data on ARV prophylaxis in infants (indicator 7) were only available from March 2018 onwards.

Interrupted time series Poisson regression models accounting for overdispersion were used to estimate the mean value per month for each indicator in each of the two time periods, as well as the trend over time in each time period, and the relative drop at the start of the pandemic [12]. Where there was evidence (p<0.05) of a trend over time prior to COVID-19, estimates of the mean value pre-COVID-19 were restricted to a six-month window excluding the doctor's strike (May-August 2019 and February-March 2020), and estimates for the whole time period were presented as a sensitivity analysis.

**Table 1. Definition of indicators.**

| Indicator | | | Programme/ population | Numerator | | Denominator | |
|---|---|---|---|---|---|---|---|
| | | | | Definition | Source | Definition | Source |
| Antenatal care | 1 | Estimated proportion of women booking for ANC | Population | Number of pregnant women booking for ANC | DHIS-2 [P1] | Estimated number of pregnant women | ZIMSTAT |
| | 2 | Estimated proportion of deliveries in a healthcare facility | Population | Number of institutional deliveries | DHIS-2 [P22] | Estimated number of pregnant women | ZIMSTAT |
| Maternal HIV care in antenatal care | 3 | Estimated proportion of pregnant women tested once for HIV in ANC | Programme | Number of pregnant women receiving at least one test for HIV | DHIS-2 [P3] | Number of pregnant women booking for ANC, excluding those presenting HIV-positive | DHIS-2 [P1-P2] |
| | | | Population | Number of pregnant women tested for HIV | DHIS-2 [P3] | Estimated number of pregnant women | ZIMSTAT |
| | 4 | Estimated proportion of pregnant women subsequently retested for HIV in ANC | Programme | Number of pregnant women retested for HIV after initial negative test | DHIS-2 [P5] | Number of pregnant women in ANC, excluding those known to be HIV-positive (at booking or on first test) | DHIS-2 [P1-P2-P4] |
| | | | Population | Number of pregnant women retested for HIV | DHIS-2 [P5] | Estimated number of pregnant women | ZIMSTAT |
| | 5 | Estimated proportion of pregnant women with HIV on ART | Programme | Number of pregnant women on ART | DHIS-2 [P9+P10 +P11] | Number of pregnant women presenting or testing positive for HIV in ANC | DHIS-2 [P2 +P4+P6] |
| | | | Population | Number of pregnant women on ART | DHIS-2 [P9+P10 +P11] | Estimated number of pregnant women with HIV | Spectrum |
| | 6 | Estimated proportion of women arriving in labour/ delivery with unknown HIV status | Programme | Number of pregnant women arriving in labour/ delivery with unknown status | DHIS-2 [P16] | Number of institutional deliveries | DHIS-2 [P22] |
| Care for HIV-exposed infants | 7 | Estimated proportion of HIV-exposed infants initiating ARV prophylaxis | Programme | Number of HIV-exposed infants initiating on ARV prophylaxis | DHIS-2 [P31+P32] | Number of pregnant women with HIV or diagnosed within 24 months post-delivery | DHIS-2 [P23 +P26+P28] |
| | | | Population | Number of HIV-exposed infants initiating on ARV prophylaxis | DHIS-2 [P31+P32] | Estimated number of HIV-exposed infants born | Spectrum |
| | 8 | Estimated proportion of HIV-exposed infants receiving co-trimoxazole prophylaxis | Programme | Number of HIV-exposed initiating on co-trimoxazole prophylaxis | DHIS-2 [P33] | Number of pregnant women with HIV or diagnosed within 24 months post-delivery | DHIS-2 [P23 +P26+P28] |
| | | | Population | Number of HIV-exposed initiating on co-trimoxazole prophylaxis | DHIS-2 [P33] | Estimated number of HIV-exposed infants born | Spectrum |
| | 9 | Estimated proportion of HIV-exposed infants receiving EID | Programme | Number of HIV-exposed infants with EID sample collected | DHIS-2 [P34] | Number of pregnant women with HIV or diagnosed within 24 months post-delivery | DHIS-2 [P23 +P26+P28] |
| | | | Population | Number of HIV-exposed infants with EID sample collected | DHIS-2 [P34] | Estimated number of HIV-exposed infants born | Spectrum |
| Care for infants with HIV | 10 | Estimated proportion of infants with HIV diagnosed | Population | Number of HIV-exposed infants testing HIV-positive | DHIS-2 [P35] | Estimated number of HIV-positive infants | Spectrum |
| | 11 | Estimated proportion of infants with HIV initiating ART | Programme | Number of infants initiating ART | DHIS-2 [P36] | Number of HIV-exposed infants testing HIV-positive | DHIS-2 [P35] |
| | | | Population | Number of infants initiating ART | DHIS-2 [P36] | Estimated number of HIV-positive infants | Spectrum |

Estimated numbers of pregnant women with HIV, of HIV-exposed infants and of HIV-infected infants from Spectrum, and estimated number of pregnant women from ZIMSTAT, were estimated per year and divided by 12 to give monthly estimates

## Results

There were an estimated 2,150,323 pregnancies in the population across the study period (4 years, 9 months), of which 268,059 (12%) were in women living with HIV. There were an estimated 22,529 HIV infections in infants through mother-to-child transmission. The proportion of healthcare facilities reporting data to DHIS-2 was above 98% during early 2019, falling to 91% in December 2019 during the doctors' strike, but remained above 97% throughout 2020 and 2021 (S1 Fig). 1,949,936 ANC booking (first) visits were captured in DHIS-2 across the study period.

The mean estimated proportion of pregnant women booking for the first ANC visit per month did not change (91% pre-pandemic vs 91% during pandemic, p = 0.95), despite dropping to 47% in April 2020 (Table 2, Fig 1). The estimated proportion of women delivering in a healthcare facility was also similar in each time period (74% vs 73%, p = 0.21). The estimated proportion of women delivering with unknown HIV status was higher during the pandemic compared to prior (2.4% vs. 3.3%, p<0.001).

At a programme level, the estimated proportion of pregnant women who received at least one HIV test in ANC was lower during the pandemic (98% vs 96%, p<0.001), reflecting a drop in monthly HIV testing in April 2020 (3.6% reduction vs March 2020 (95% CI 2.2–5.1), p<0.001) with gradual recovery towards pre-pandemic levels (0.1% increase per month during COVID-19 (95% CI 0.0–0.2), p = 0.048), reaching 98% by September 2021 (Fig 2). The estimated proportion of women who were retested for HIV among those initially negative in pregnancy also fell in April 2020 (39% reduction (32–45%), p<0.001), and the subsequent increase was much slower, with the estimated proportion retested only reaching 39% by September 2021 compared to an average of 53% in the pre-pandemic period. At a population level, there was no evidence for a significant change in initial testing, but trends in retesting were similar to those at a programme level.

The mean estimated proportion of pregnant women living with HIV on ART remained unchanged at programme level (98% vs 98%, p = 0.26), but there was some evidence that it was lower during COVID-19 at population level (86% vs 80%, p = 0.049). The estimated proportion of pregnant women delivering with unknown HIV status fell over time prior to the pandemic (mean 2.4%), and subsequently increased immediately prior to the start of the pandemic, during which it was stable over time (mean 3.3%).

Postnatal antiretroviral prophylaxis coverage among HIV-exposed infants was lower during COVID-19 at programme (94% vs 87%, p = 0.001) and population levels (76% vs 68%, p<0.001) (Fig 3). The estimated proportion of HIV-exposed infants who received co-trimoxazole prophylaxis was lower during the pandemic at a population level (72% vs 65%, p = 0.002), but there was no significant difference at a programme level (87% vs 83%, p = 0.31). Despite initial decreases, overall there was no significant difference in the estimated proportion of HIV-exposed infants receiving EID pre- vs. during COVID-19 (programme: 107% vs 103%, p = 0.52; population: 87% vs 79%, p = 0.081).

The estimated proportion of infants with HIV who were diagnosed fell from 27% pre-COVID-19 to 18% during COVID-19, (p<0.001) (Fig 4). The estimated proportion of infants with HIV on ART was stable at a programme (88% vs 90%, p = 0.82) but at a population level was substantially lower and fell during COVID-19 (22% vs 16%, p = 0.004).

Results are presented as cascades of care for both women and infants living with HIV in Fig 5. Prior to the pandemic, 88% of women living with HIV were attending ANC and diagnosed with HIV, and 86% were on ART, falling to 82% and 80% respectively during the pandemic. Prior to the pandemic, 27% of infants living with HIV were diagnosed, and 31% had started ART; during the pandemic these estimated proportions fell to 18% and 16% respectively.

**Table 2. Comparison of indicators pre-COVID-19 and during COVID-19.**

| Indicator | | | Programme/ population | Pre-COVID-19 | During-COVID-19 | p | Change per month pre-COVID-19 | Change at start of pandemic** | Change per month during-COVID-19 | Proportion in September 2021 |
|---|---|---|---|---|---|---|---|---|---|---|
| | | | | Estimate (95% CI) | | | Incidence rate ratio (95% CI), p-value | | | |
| Antenatal care | 1 | Estimated proportion of pregnant women booking for ANC | Population | 91% (87%, 94%) | 91% (86%, 96%) | 0.95 | 1.00 (1.00, 1.00), p = 0.95 | 0.99 (0.86, 1.13), p = 0.85 | 1.00 (0.99, 1.01), p = 0.69 | 93% |
| | 2 | Estimated proportion of women delivering in healthcare facility | Population | 74% (73%, 76%) | 73% (71%, 75%) | 0.21 | 1.00 (1.00, 1.00), p = 0.79 | 0.89 (0.84, 0.95), p<0.001 | 1.01 (1.01, 1.02), p<0.001 | 85% |
| Maternal HIV care in antenatal care | 3 | Estimated proportion of pregnant women tested for HIV | Programme | 98% (98%, 99%) | 96% (95%, 96%) | <0.001 | 1.00 (1.00, 1.00), p = 0.67 | 0.96 (0.94, 0.98), p<0.001 | 1.00 (1.00, 1.00), p = 0.048 | 98% |
| | | | Population | 81% (78%, 85%) | 80% (76%, 84%) | 0.60 | 1.00 (1.00, 1.00), p = 0.92 | 0.95 (0.83, 1.09), p = 0.51 | 1.00 (0.99, 1.01), p = 0.46 | 83% |
| | 4 | Estimated proportion of pregnant women retested for HIV | Programme | 53% (51%, 55%) | 36% (34%, 38%) | <0.001 | 1.00 (1.00, 1.00), p = 0.36 | 0.61 (0.55, 0.68), p<0.001 | 1.01 (1.01, 1.02), p<0.001 | 39% |
| | | | Population | 42% (41%, 43%) | 29% (28%. 31%) | <0.001 | 1.00 (1.00, 1.00), p = 0.61 | 0.61 (0.54, 0.69), p<0.001 | 1.02 (1.01, 1.03), p = 0.001 | 32% |
| | 5 | Estimated proportion of pregnant women with HIV on ART | Programme | 98% (98%, 99%) | 98% (97%, 98%) | 0.26 | 1.00 (1.00, 1.00), p = 0.39 | 1.00 (0.98, 1.02), p = 0.87 | 1.00 (1.00, 1.00), p = 0.19 | 97% |
| | | | Population | 86% (83%, 89%) | 80% (76%, 85%) | 0.049 | 1.00 (1.00, 1.00), p = 0.79 | 0.93 (0.81, 1.07), p = 0.32 | 1.00 (0.99, 1.01), p = 0.89 | 83% |
| | 6 | Estimated proportion of women delivering with unknown HIV status* | Programme | 2.4% (2.2%, 2.6%) | 3.3% (3.2%, 3.5%) | <0.001 | 1.01 (0.99, 1.03), p = 0.20 | 1.37 (1.17, 1.60), p<0.001 | 0.99 (0.99, 1.00), p = 0.042 | 2.7% |
| Care for HIV-exposed infants | 7 | proportion of HIV-exposed infants initiating ARV prophylaxis | Programme | 94% (91%, 96%) | 87% (84%, 90%) | 0.001 | 1.00 (1.00, 1.00), p = 0.73 | 1.01 (0.93, 1.10), p = 0.76 | 0.99 (0.99, 1.00), p = 0.004 | 72% |
| | | | Population | 76% (73%, 78%) | 68% (65%, 70%) | <0.001 | 1.00 (1.00, 1.01), p = 0.51 | 0.85 (0.76, 0.94), p = 0.001 | 1.00 (1.00, 1.01), p = 0.34 | 69% |
| | 8 | Estimated proportion of HIV-exposed infants receiving CTX* | Programme | 87% (81%, 95%) | 83% (79%. 88%) | 0.31 | 1.00 (0.98, 1.02), p = 0.85 | 1.01 (0.85, 1.19), p = 0.95 | 0.99 (0.98, 1.00), p = 0.11 | 68% |
| | | | Population | 72% (68%, 77%) | 65% (62%, 67%) | 0.002 | 1.00 (0.98, 1.01), p = 0.56 | 0.88 (0.77, 1.01), p = 0.067 | 1.00 (1.00, 1.01), p = 0.22 | 64% |
| | 9 | Estimated proportion of HIV-exposed infants receiving EID | Programme | 107% (104%, 110%) | 103% (93%, 114%) | 0.52 | 1.00 (1.00, 1.00), p = 0.054 | 0.75 (0.65, 0.87), p<0.001 | 1.22 (1.10, 1.35), p<0.001 | - |
| | | | Population | 87% (84%, 89%) | 79% (71%, 87%) | 0.081 | 1.00 (1.00, 1.00), p = 0.151 | 0.73 (0.62, 0.87), p<0.001 | 1.18 (1.05, 1.33), p = 0.005 | - |
| Care for infants with HIV | 10 | Estimated proportion of infants with HIV diagnosed | Population | 27% (25%, 28%) | 18% (17%, 20%) | <0.001 | 1.00 (1.00, 1.00), p = 0.29 | 0.71 (0.55, 0.90), p = 0.005 | 1.01 (0.99, 1.03), p = 0.58 | 27% |
| | 11 | Estimated proportion of infants with HIV on ART* | Programme | 88% (78%, 100%) | 90% (83%, 97%) | 0.82 | 0.99 (0.96, 1.03), p = 0.72 | 1.11 (0.82, 1.49), p = 0.51 | 0.99 (0.98, 1.01), p = 0.52 | 52% |
| | | | Population | 22% (19%, 26%) | 16% (15%, 18%) | 0.004 | 0.95 (0.92, 1.00), p = 0.028 | 0.99 (0.69, 1.42), p = 0.96 | 1.00 (0.98, 1.02), p = 0.99 | 14% |

*Indicator was significantly increasing/decreasing over time pre-COVID-19; pre-COVID-19 estimates (both proportion and change per month) have been restricted to a 6 month window (March-August 2019). Results using all data since January 2017 are presented in S2 Table.

**This estimate is based on the difference in the point estimates for March and April 2020 from the interrupted time series model.

## (1) Estimated proportion of pregnant women booking for antenatal care (pre=91%, during=91%, p=0.95)

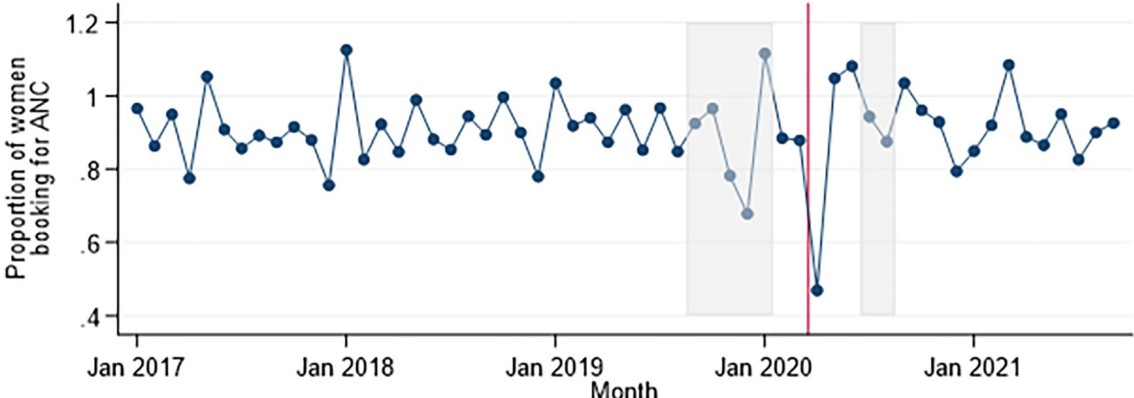

## (2) Estimated proportion of women delivering in a healthcare facility (pre=74%, during=73%, p=0.21)

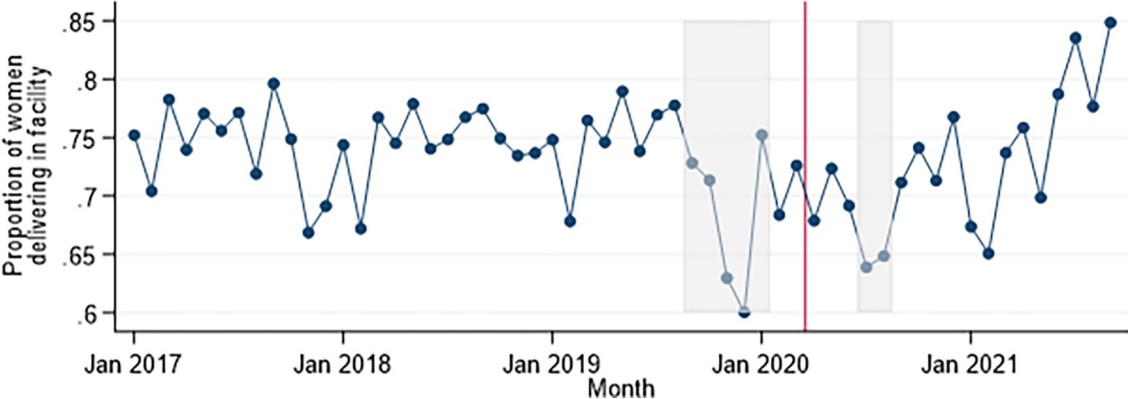

**Fig 1. Indicators relating to antenatal care over time: (1) estimated proportion of pregnant women booking for antenatal care; and (2) estimated proportion of women delivering in a healthcare facility.** Grey boxes represent the doctors' and nurses' strikes, and the red line represents the start of the COVID-19 pandemic in Zimbabwe.

Results of the sensitivity analyses around the inclusion/exclusion of time periods during the healthcare worker strikes were similar to the main analysis (S1 Table). Results were generally consistent across provinces (S2–S12 Figs).

## Discussion

In this analysis we explored the impact of the COVID-19 pandemic on the provision and uptake of PMTCT services and infant diagnosis and treatment in Zimbabwe. Despite an initial drop in April 2020, overall there was no impact on the estimated proportion of pregnant women having at least one ANC visit. However testing of women for HIV fell substantially. Although the estimated proportion of women receiving at least one test gradually returned to normal, the estimated proportion of women retested (following an initial negative result) was not yet at pre-pandemic levels by September 2021. At a programme level, there was no impact

(3) Estimated proportion of pregnant women tested once for HIV (programme: pre=98%, during=96%, p<0.001; population: pre=81%, during=80%, p=0.60)

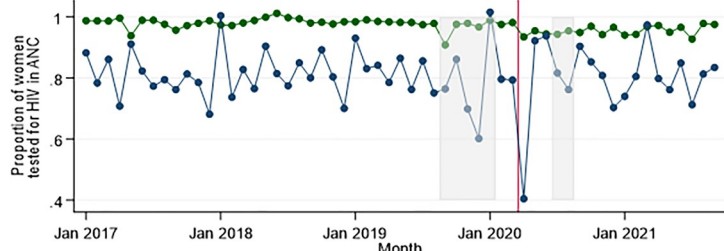

(4) Estimated proportion of pregnant women receiving a subsequent test for HIV (programme: pre=53%, during=36%, p<0.001; population: pre=42%, during=29%, p<0.001)

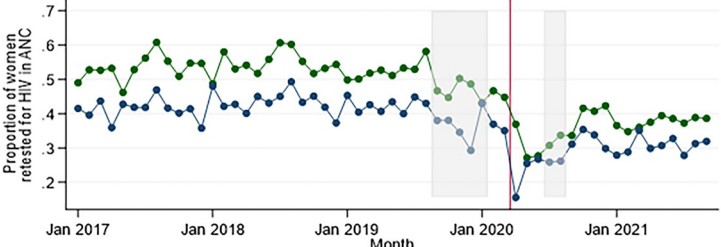

(5) Estimated proportion of pregnant women with HIV on ART (programme: pre=98%, during=98%, p=0.26; population: pre=86%, during=80%, p=0.049)

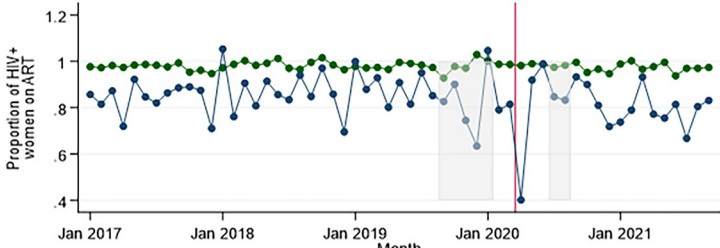

(6) Estimated proportion of pregnant women arriving in labour/delivery with unknown HIV status (pre=2.4%, during=3.3%, p<0.001)

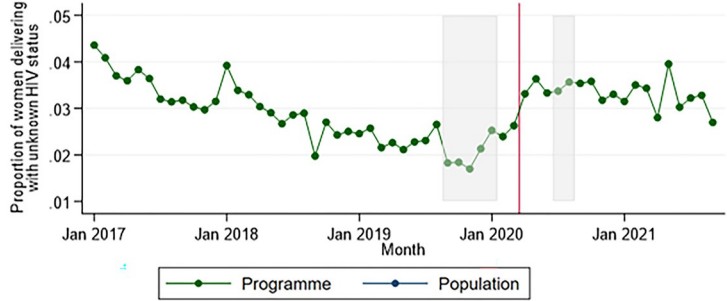

**Fig 2.** Indicators relating to maternal HIV care over time: (3) estimated proportion of pregnant women tested once for HIV; (4) estimated proportion of pregnant women receiving a subsequent test for HIV; (5) estimated proportion of pregnant women with HIV on ART; (6) estimated proportion of pregnant women arriving in labour/delivery with unknown HIV status. Grey boxes represent the doctors' and nurses' strikes, and the red line represents the start of the COVID-19 pandemic in Zimbabwe.

of the pandemic on ART coverage among pregnant women with HIV. However, there were falls at a population level, that is among the total population including undiagnosed women, likely because of reduced testing of women in pregnancy. In infants, although testing of HIV-exposed infants remained stable, postnatal prophylaxis coverage was lower at both programme

(7) Estimated proportion initiating ARV prophylaxis (programme: pre=94%, during=87%, p<0.001; population: pre=76%, during=68%, p<0.001)

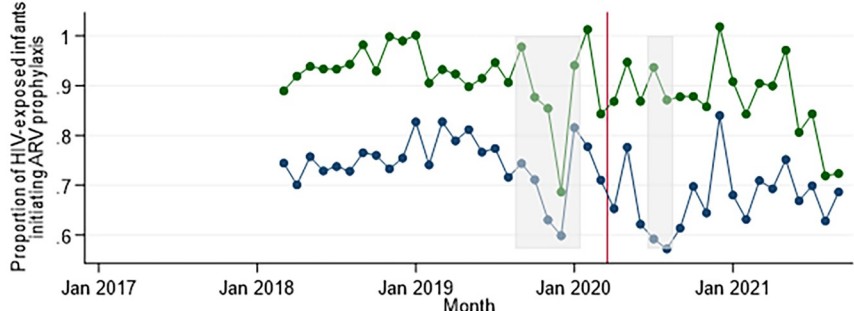

(8) Estimated proportion receiving co-trimoxazole prophylaxis (programme: pre=87%, during=83%, p=0.31; population: pre=72%, during=65%, p=0.002)

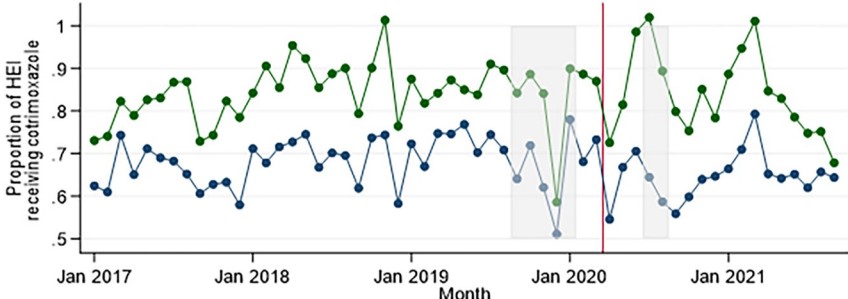

(9) Estimated proportion receiving early infant diagnosis (programme: pre=107%, during=103%, p=0.52; population: pre=87%, during=79%, p=0.081)

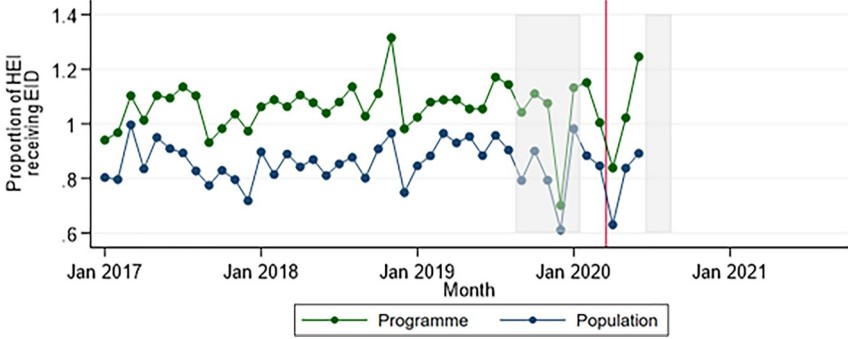

**Fig 3.** Indicators relating to care for HIV-exposed infants over time: (7) estimated proportion initiating ARV prophylaxis; (8) estimated proportion receiving co-trimoxazole prophylaxis; (9) estimated proportion receiving early infant diagnosis (EID). Grey boxes represent the doctors' and nurses' strikes, and the red line represents the start of the COVID-19 pandemic in Zimbabwe.

and population levels, furthermore, the estimated proportion of infants with HIV on ART was extremely low at a population level, worsened by the COVID-19 pandemic.

We observed a drop in first ANC booking visits early in the pandemic, however this subsequently returned to normal levels, and indeed when comparing overall proportions of women booking for ANC before and during the pandemic there was no change, suggesting women

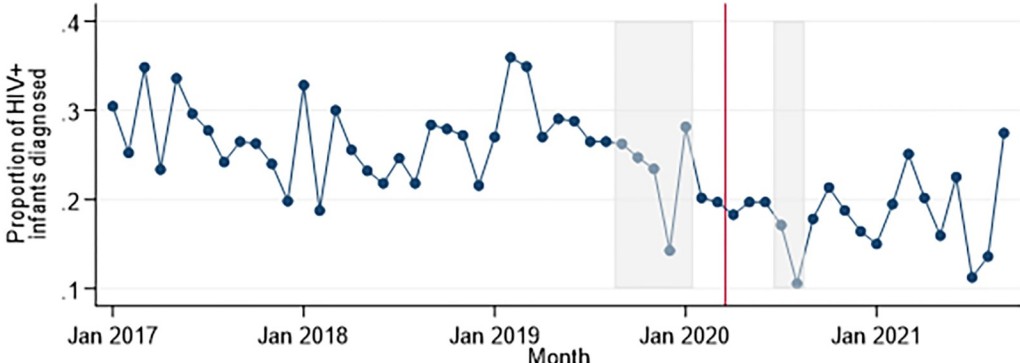

(10) Estimated proportion of infants with HIV diagnosed (pre=27%, during=18%, p<0.001)

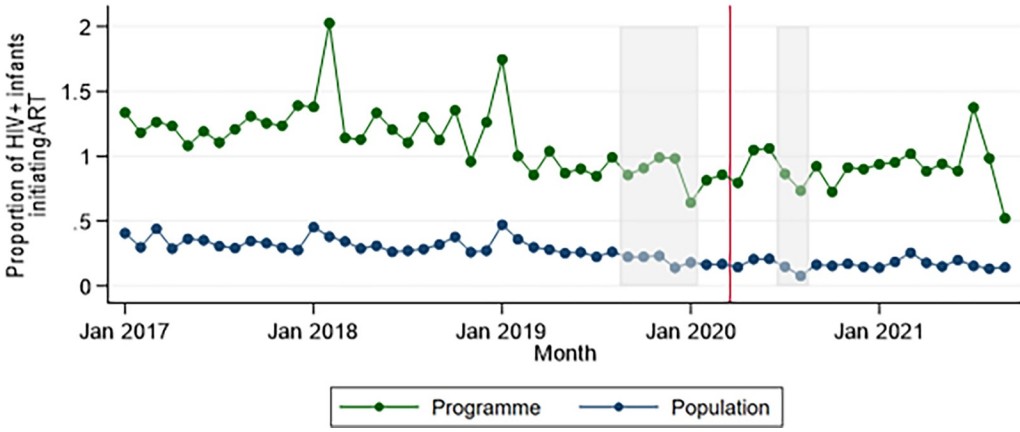

(11) Estimated proportion of infants with HIV initiating ART (programme: pre=88%, during=90%, p=0.82; population: pre=22%, during=16%, p=0.004)

**Fig 4. Indicators relating to care for infants with HIV over time: (10) estimated proportion of infants with HIV diagnosed; (11) estimated proportion of infants with HIV initiating ART.** Grey boxes represent the doctors' and nurses' strikes, and the red line represents the start of the COVID-19 pandemic in Zimbabwe.

who had initially been missed due to early disruption did eventually book for care. Studies in other countries, including Uganda, Haiti, Sierra Leone, have reported similar early falls in ANC booking which subsequently stabilised [13, 14].

The reduction in HIV testing during pregnancy during the pandemic may be partly explained by reported issues with supply of test kits during this time [15]. In particular, we observed a drop in retesting of women who tested negative early in pregnancy, with national guidelines recommending repeat HIV testing during the third trimester, important given that seroconversion during pregnancy is a key risk factor for transmission [16]. Prior to the pandemic, an average of 2000 women were diagnosed at retesting. Results from a concurrent qualitative study as part of the same project reported that early in the pandemic some women were told at ANC booking not to return to ANC until they were in labour, and that even following this the recommended number of ANC visits was lower than normal, resulting in reduced opportunities for testing [17]. Fewer women being tested presumably led to fewer women being diagnosed with HIV, which likely led to drops at a population level of indicators later down the cascade of care, including for infants, although these generally did not change at

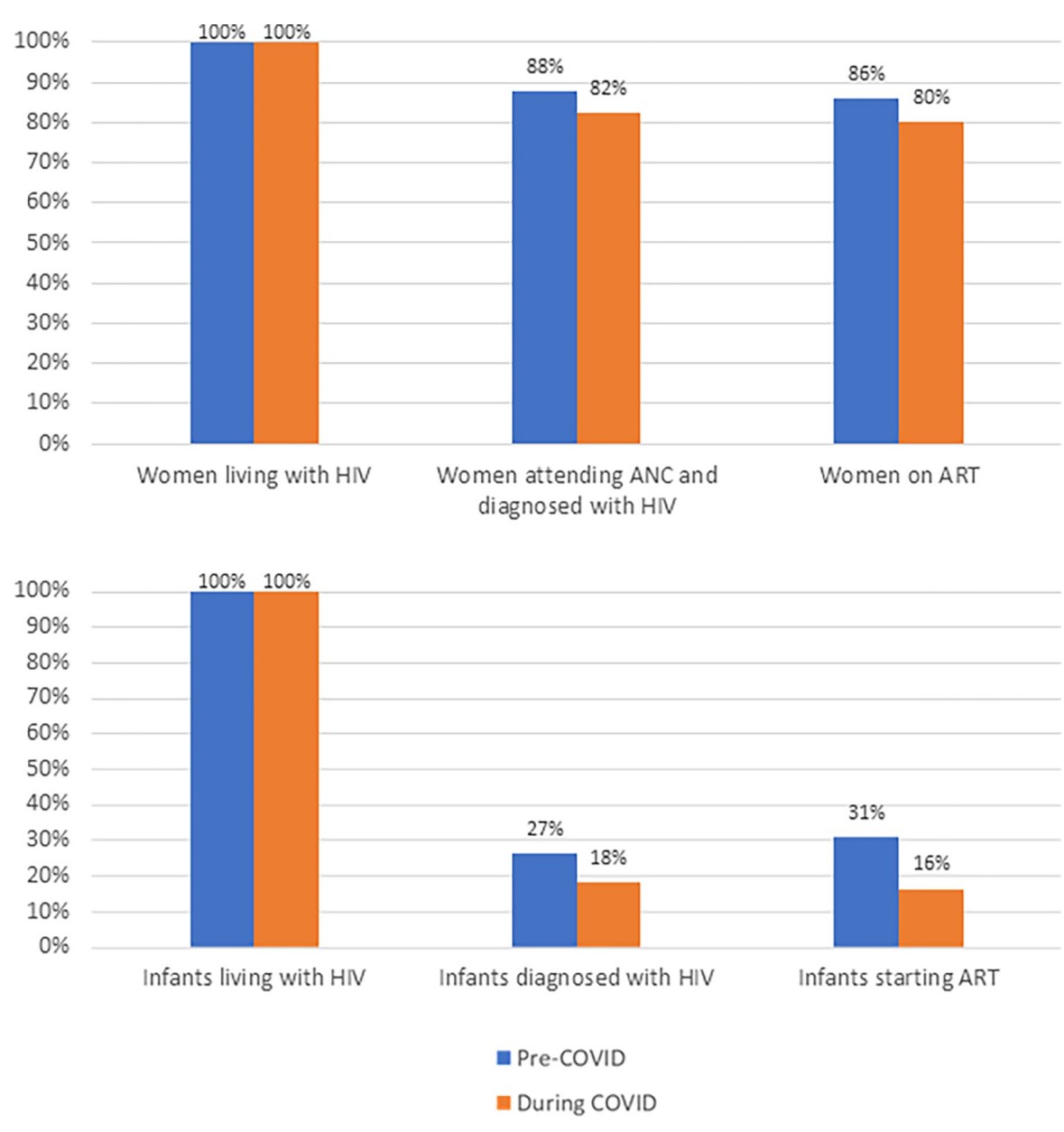

**Fig 5. Cascades of care over time.**

programme level. Results here are broadly consistent with those from a study of data from 12 countries in sub-Saharan Africa (including Zimbabwe), which reported population-level decreases in key PMTCT indicators [18].

We observed an increase in the estimated proportion of deliveries which occurred at a facility part way through the pandemic. This may have been a consequence of an increase in pregnancies among adolescents during the pandemic, which has been anecdotally reported, who are more likely to deliver in a facility. This is in contrast to results of an analysis of data from 37 healthcare facilities in 6 low and middle income countries, which reported declines in facility-based deliveries from March to December 2020 in all countries but one [14].

Aside from maternal HIV testing, there was generally little impact of the pandemic on indicators among patients in HIV care (programme level). In part this is likely a result of

healthcare worker commitment, and national guidelines developed to prioritise key services, which ensured continuation of services. However, we did observe drops in many indicators at a population level; this may have been a result of fewer pregnant women being tested and diagnosed with HIV, resulting in fewer women and infants in PMTCT services. For example, Although approximately 90% of infants with HIV engaged in programmatic care were receiving ART, at a population level there are extremely low rates of infants with HIV receiving ART. This reflects global estimates of ART coverage which demonstrate a widening gap in coverage between children and adults [19], and is concerning given the high mortality rates in the first 2 years of life [20] and the known benefits of early ART for later outcomes [21]. Similarly, at a population level, we observed a reduction in the estimated proportion of infants with HIV who were diagnosed, which was low prior to COVID-19, suggesting that even more children with HIV were being missed (with 1 in 4 being appropriately diagnosed pre-pandemic, falling to 1 in 5). It is possible that Spectrum estimates used for population-level denominators did not accurately reflect changes during the pandemic, in particular since 2021 estimates were unavailable at the time of analysis and so 2020 estimates were again used. We also observed falls at a programme level as both postnatal antiretroviral prophylaxis and co-trimoxazole were lower during the COVID-19 pandemic. The impact was not immediately after the start of the pandemic but began in mid-2021; the reasons for this are unclear.

The strengths of the analysis include the use of data for all facilities in Zimbabwe which provide ANC and the inclusion of indicators across the PMTCT cascade. This analysis has several limitations. Firstly, we used aggregate data rather than individual patient-level data, resulting in estimates over 100% for some months. Similarly, for some population level indicators, denominators were the total population rather than the population of interest; for example, the population-level denominator for maternal HIV diagnosis did not exclude women already known to be living with HIV, in the same way the programme-level denominator did. However, since the focus of this analysis was exploring trends over time, rather than absolute estimates, the estimates upon which our main conclusions are based should be valid. Secondly, ZIMSTAT and Spectrum estimates are calculated per year, and we then scaled these to get the corresponding monthly estimates, assuming a constant annual birth rate which may not be correct.

In conclusion, results from this study demonstrate a drop in HIV testing of women in ANC during the COVID-19 pandemic, which likely impacted at a population level the proportion of pregnant women living with HIV who were started on ART, and similarly exacerbated the already low proportion of HIV-exposed infants diagnosed and on treatment, despite little change to care at a programmatic level. COVID-19 challenged healthcare services in Zimbabwe, which were already overstretched prior to the pandemic [22]. Beyond HIV, there was evidence of an impact of the pandemic on other health indicators in Zimbabwe, for example an increase in adverse birth outcomes including still births and neonatal deaths [23]. Further research is required to better understand the reasons for the declines in maternal HIV testing reported here, in order to identify weaknesses in health services and to better prepare for subsequent waves of COVID-19, as well as other future pandemics, and to ensure Zimbabwe stays on track to eliminate MTCT by 2025.

## Supporting information

**S1 Fig. Proportion of healthcare facilities reporting to DHIS-2 over time.**
(TIF)

**S2 Fig. Estimated proportion of pregnant women booking for antenatal care over time, by province.**
(TIF)

**S3 Fig. Estimated proportion of women delivering in a healthcare facility over time, by province.**
(TIF)

**S4 Fig. Estimated proportion of pregnant women tested once for HIV over time, by province.**
(TIF)

**S5 Fig. Estimated proportion of pregnant women receiving a subsequent test for HIV over time, by province.**
(TIF)

**S6 Fig. Estimated proportion of pregnant women with HIV on ART over time, by province.**
(TIF)

**S7 Fig. Estimated proportion of pregnant women arriving in labour/delivery with unknown HIV status over time, by province.**
(TIF)

**S8 Fig. Estimated proportion of HIV-exposed infants initiating ARV prophylaxis over time, by province.**
(TIF)

**S9 Fig. Estimated proportion of HIV-exposed infants initiating cotrimoxazole prophylaxis over time, by province.**
(TIF)

**S10 Fig. Estimated proportion of HIV-exposed infants receiving early infant diagnosis (EID) over time, by province.**
(TIF)

**S11 Fig. Estimated proportion of infants with HIV diagnosed over time, by province.**
(TIF)

**S12 Fig. Estimated proportion of infants with HIV initiating ART over time, by province.**
(TIF)

**S1 Table. Results of sensitivity analyses.**
(DOCX)

**S2 Table. Results using data from the full time period for indicators where there was an ongoing trend prior to the start of the pandemic.**
(DOCX)

**S1 File. DHIS-2 monthly summary return form.**
(PDF)

**S1 Data. Dataset.**
(XLSX)

## Author Contributions

**Conceptualization:** Elizabeth Chappell, Diana M. Gibb, Deborah Ford, Angela Mushavi, Mutsa Bwakura-Dangarembizi.

**Data curation:** Anesu Chimwaza, Ngoni Manika, Angela Mushavi.

**Formal analysis:** Elizabeth Chappell, Deborah Ford.

**Funding acquisition:** Elizabeth Chappell, Catherine J. Wedderburn, Zivai Mupambireyi Nenguke, Hannah Gannon, Frances Cowan, Tom Gibb, Michelle Heys, Felicity Fitzgerald, Andrew Phillips, Simbarashe Chimhuya, Diana M. Gibb, Deborah Ford, Angela Mushavi, Mutsa Bwakura-Dangarembizi.

**Methodology:** Elizabeth Chappell, Deborah Ford.

**Project administration:** Elizabeth Chappell, Mutsa Bwakura-Dangarembizi.

**Supervision:** Angela Mushavi, Mutsa Bwakura-Dangarembizi.

**Writing – original draft:** Elizabeth Chappell.

**Writing – review & editing:** Elizabeth Chappell, Anesu Chimwaza, Ngoni Manika, Catherine J. Wedderburn, Zivai Mupambireyi Nenguke, Hannah Gannon, Frances Cowan, Tom Gibb, Michelle Heys, Felicity Fitzgerald, Andrew Phillips, Simbarashe Chimhuya, Diana M. Gibb, Deborah Ford, Angela Mushavi, Mutsa Bwakura-Dangarembizi.

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
