## [Decision Letter · Decision Letter 0]

31 May 2023

PGPH-D-22-02047

Impact of the COVID-19 pandemic on the provision and uptake of services for the prevention of mother-to-child transmission of HIV in Zimbabwe

Dear Dr. Chappell,

Thank you for submitting your manuscript to PLOS Global Public Health. After careful consideration, we feel that it has merit but does not fully meet PLOS Global Public Health’s publication criteria as it currently stands. Therefore, we invite you to submit a revised version of the manuscript that addresses the points raised during the review process.

The review comments can be found at the end of this email, together with any comments from the Editorial Office regarding formatting changes or additional information required to meet the journal's policies at this time.

Please note that your revision may be subject to further review and that this initial decision does not guarantee acceptance. 

Please submit your revised manuscript by  30-June-2023. If you will need more time than this to complete your revisions, please reply to this message or contact the journal office at globalpubhealth@plos.org. Please include the following items when submitting your revised manuscript:

We look forward to receiving your revised manuscript.

Kind regards,

Mbuzeleni Hlongwa, Ph.D

Academic Editor

Journal Requirements:

b. State what role the funders took in the study. If the funders had no role in your study, lease state: “The funders had no role in study design, data collection and analysis, decision to publish, or preparation of the manuscript.”

c. If any authors received a salary from any of your funders, please state which authors and which funders

Additional Editor Comments (if provided):

Reviewers' comments:

Reviewer's Responses to Questions

**Comments to the Author**

1. Does this manuscript meet PLOS Global Public Health’s publication criteria? Is the manuscript technically sound, and do the data support the conclusions? The manuscript must describe methodologically and ethically rigorous research with conclusions that are appropriately drawn based on the data presented.

Reviewer #1: Yes

Reviewer #2: Yes

2. Has the statistical analysis been performed appropriately and rigorously?

Reviewer #1: Yes

Reviewer #2: Yes

3. Have the authors made all data underlying the findings in their manuscript fully available (please refer to the Data Availability Statement at the start of the manuscript PDF file)?

Reviewer #1: Yes

Reviewer #2: Yes

4. Is the manuscript presented in an intelligible fashion and written in standard English?

Reviewer #1: Yes

Reviewer #2: Yes

5. Review Comments to the Author

Reviewer #1: The authors present a Impact of the COVID-19 pandemic on the provision and uptake of services for the

prevention of mother-to-child transmission of HIV in Zimbabwe. The paper is well written, informative, novel and contains plenty of data. This reviewer is in favor of accepting this manuscript for publication

Reviewer #2: Excellent paper.

Please make a few minor edits:

1. In Paragraph 1 of the introduction, use past instead of present tense where appropriate

2. In Table 1 - for denominators, please use "estimated number" instead of "number" where appropriate

3. Please do the same for the results narrative as well - be very clear in regards to what are "estimated number of" and "estimated proportion of" vs. absolute counts.

6. PLOS authors have the option to publish the peer review history of their article (what does this mean?). If published, this will include your full peer review and any attached files.

**Do you want your identity to be public for this peer review?** For information about this choice, including consent withdrawal, please see our Privacy Policy.

Reviewer #1: No

Reviewer #2: **Yes: **Beth A Tippett Barr

<quillbot-extension-portal></quillbot-extension-portal>

---

## [Decision Letter · Decision Letter 1]

24 Jul 2023

Impact of the COVID-19 pandemic on the provision and uptake of services for the prevention of mother-to-child transmission of HIV in Zimbabwe

PGPH-D-22-02047R1

Dear Dr Chappell,

We are pleased to inform you that your manuscript 'Impact of the COVID-19 pandemic on the provision and uptake of services for the prevention of mother-to-child transmission of HIV in Zimbabwe' has been provisionally accepted for publication in PLOS Global Public Health.

Best regards,

Julia Robinson

Executive Editor

Reviewer Comments (if any, and for reference):

Reviewer's Responses to Questions

**Comments to the Author**

1. If the authors have adequately addressed your comments raised in a previous round of review and you feel that this manuscript is now acceptable for publication, you may indicate that here to bypass the “Comments to the Author” section, enter your conflict of interest statement in the “Confidential to Editor” section, and submit your "Accept" recommendation.

Reviewer #2: All comments have been addressed

2. Does this manuscript meet PLOS Global Public Health’s publication criteria? Is the manuscript technically sound, and do the data support the conclusions? The manuscript must describe methodologically and ethically rigorous research with conclusions that are appropriately drawn based on the data presented.

Reviewer #2: Yes

3. Has the statistical analysis been performed appropriately and rigorously?

Reviewer #2: Yes

4. Have the authors made all data underlying the findings in their manuscript fully available (please refer to the Data Availability Statement at the start of the manuscript PDF file)?

Reviewer #2: Yes

5. Is the manuscript presented in an intelligible fashion and written in standard English?

Reviewer #2: Yes

6. Review Comments to the Author

Reviewer #2: Thank you for addressing the previous comments.

7. PLOS authors have the option to publish the peer review history of their article (what does this mean?). If published, this will include your full peer review and any attached files.

**Do you want your identity to be public for this peer review?** For information about this choice, including consent withdrawal, please see our Privacy Policy.

Reviewer #2: **Yes: **Beth A Tippett Barr
